# Quasi-Continuous Network Structure Greatly Improved the Anti-Arc-Erosion Capability of Ag/Y_2_O_3_ Electrical Contacts

**DOI:** 10.3390/ma15072450

**Published:** 2022-03-26

**Authors:** Rui Yang, Shaohong Liu, Hao Cui, Hongwei Yang, Yiming Zeng, Manmen Liu, Jialin Chen, Ming Wen, Wei Wang, Zhengtang Luo, Xudong Sun

**Affiliations:** 1Key Laboratory for Anisotropy and Texture of Materials (Ministry of Education), School of Materials Science and Engineering, Northeastern University, Shenyang 110819, China; yangrui@stumail.neu.edu.cn (R.Y.); wangw@atm.neu.edu.cn (W.W.); 2State Key Laboratory of Advanced Technologies for Comprehensive Utilization of Platinum Metals, Kunming Institute of Precious Metals, Kunming 650106, China; cuihao@ipm.com.cn (H.C.); nanolab@ipm.com.cn (H.Y.); zengym0871@126.com (Y.Z.); lmm@ipm.com.cn (M.L.); cjl@ipm.com.cn (J.C.); wen@ipm.com.cn (M.W.); 3Department of Chemical and Biological Engineering, William Mong Institute of Nano Science and Technology, The Hong Kong University of Science and Technology, Hong Kong 999077, China; keztluo@ust.hk; 4Foshan Graduate School, Northeastern University, Foshan 528311, China

**Keywords:** electrical contact, arc erosion, contact resistance, silver, Y_2_O_3_

## Abstract

Ag/Y_2_O_3_ has excellent potential to replace Ag/CdO as the environmentally friendly electrical contact material. Using spherical Y_2_O_3_ as the starting material, Ag/Y_2_O_3_ contacts with a quasi-continuous network structure were successfully fabricated by a low-energy ball milling treatment. The mean size of Y_2_O_3_ used ranged from 243 to 980 nm. Due to the differences in the size of Y_2_O_3_, Ag/Y_2_O_3_ contacts had different primitive microstructures, thereby exhibiting distinctive anti-arc-erosion capabilities. Ag/Y_2_O_3_ contact prepared using 243 nm Y_2_O_3_ showed the best anti-arc-erosion capability and the most outstanding electrical performance measures, such as low contact resistance, less mass transfer, and no failure up to 10^5^ cycle times. The quasi-continuous network structure formed in the micro-scale was responsible for the excellent electrical performance. The short distance between Y_2_O_3_ particles in the network promoted the cathode arc motion, and thus alleviated the localized erosion. The results obtained herein may inspire further attempts to design electrical contacts rationally.

## 1. Introduction

Silver/cadmium oxide (Ag/CdO) has been the most favored electrical contact material in low-voltage electrical apparatuses since the 1960s [1,2]. Decomposition and evaporation of CdO could significantly resist the welding and erosion of the Ag matrix under multiple arcs [3]. However, cadmium is hazardous and banned in many countries [4]. The development of cadmium-free electrical contact materials has always been a worldwide issue. New materials, such as Ag/SnO_2_ [5,6], Ag/CuO [7,8], Ag/ZnO [9], Ag/MAX [10,11], Ag/TiB_2_ [12], and so on, have replaced Ag/CdO to some extent, but still face many problems, such as high and unstable contact resistance and serious ablation loss [13]. Therefore, designing and preparing new, environmentally friendly, high-performance electrical contacts has always been challenging.

Non-toxic yttrium oxide (Y_2_O_3_) has several advantages, which can be incorporated into the Ag matrix to form Ag/Y_2_O_3_ electrical contacts. Rare earth elements can promote recrystallization, refine the grains, and strengthen the matrix [14,15]. Y element’s high chemical activity [16] could help silver resist corrosion and maintain high electrical conductivity. Besides, Y_2_O_3_ has a high melting point (~2410 °C) and excellent thermal stability, thereby having outstanding high-temperature tolerance [17,18]. Fu et al. found that the addition of Y_2_O_3_ significantly enhances the wettability and machinability of Ag/SnO_2_ contacts [19]. Zhen et al. reported that yttrium-reinforced copper composites exhibit high hardness and low contact resistance [20]. In addition, our previous work showed that Ag/Y_2_O_3_ composites have high breakdown strength and less silver splashing under high-voltage spark [21]. The research on Ag/Y_2_O_3_ electrical contacts still draws enormous attention.

In addition to the composition, the microstructure also strongly influences the performance of electrical contacts. Zhang et al. found that reticulate graphene distributed in the Cu matrix endows the composite with high interfacial shear stress, thermal conductivity, and electrical conductivity [22,23]. Lin et al. showed that Ni networks could restrain the metal-pool flow and slow down the silver splashing [24,25]. Wang et al. indicated that a CuO skeleton in Ag/CuO contacts helps decrease the mass loss and prevent oxide aggregation in the eroded zone [26]. In all, the network structure could improve the composites’ mechanical strength, thermal conductivity, electrical conductivity, and anti-arc-erosion capability.

To date, published studies have provided outcomes on process parameters, microstructures, and mechanical properties of Y_2_O_3_-reinforced Al-based [27,28], Cu-based [29], and Ag-based [30] composites prepared by powder metallurgy. However, few works have been found on the electrical contact performance of Ag/Y_2_O_3_ contacts. This work aimed to prepare Ag/Y_2_O_3_ electrical contacts using spherical Y_2_O_3_ as the starting material and investigate the relationships between Y_2_O_3_ particle size, microstructure, and electrical contact performance. The phase, microstructure, and physical properties were characterized. Ag/Y_2_O_3_ contacts’ electrical contact performance was studied based on contact resistance, mass change, and the final eroded morphology. In addition, one-time eroded morphology was provided for exploring the size-effect mechanism. Research results are essential for designing and fabricating new Cd-free electrical contact materials and getting in-depth insights into the arc erosion mechanism.

## 2. Materials and Methods

### 2.1. Raw Materials

Ag powder (99.99% purity, mean size: 1 μm) was obtained from Kunming Sino-Platinum Metals Co., Ltd. (Kunming, China). Y(NO_3_)_3_∙6H_2_O (99.99%), urea (99%), ethanol (99.8%), and octadecanoic acid (99.99%) were purchased from Shanghai Sinopharm Group Co., Ltd. (Shanghai, China). All reagents were used without further purification.

### 2.2. Synthesis of Spherical Y_2_O_3_ Powders with Different Sizes

Two spherical Y_2_O_3_ precursors were prepared by forcing urea hydrolysis in Y(NO_3_)_3_ solution (0.015 mol/L) at 90 °C for 2 h. The molar ratios of Y(NO_3_)_3_∙6H_2_O to urea were 1:133 and 1:33, respectively. The precursor particles were centrifugally separated and washed with deionized water and ethanol. After annealing at 800 °C for 2 h, the precursors shift to spherical Y_2_O_3_ powders of particle size in between 200 and 400 nm.

To synthesize two larger-size Y_2_O_3_ spheres, the as-prepared spherical precursors were used as the seeds. First, 1 g seeds was dispersed in a 1 L aqueous solution, which contained Y(NO_3_)_3_ (0.015 mol/L) and urea (0.5 mol/L). Then, the mixture was heated to 90 °C and held for 2 h to obtain new precursors. After the same treatment as the seeds, the new precursors turned into Y_2_O_3_ spheres of particle sizes in between 700 and 1000 nm.

### 2.3. Fabrication of Ag/Y_2_O_3_ Sintered Compacts

Figure 1 shows the route to prepare Ag/Y_2_O_3_ sintered compact. Ag and spherical Y_2_O_3_ powders were mixed in ethanol under ultrasonic agitation. The mass ratio of Y_2_O_3_ to Ag was 9.9%. After drying the ethanol, the obtained powder was mixed in a mortar using 5 wt.‰ octadecanoic acids as a process control agent. Then, to improve the mixing and avoid the breakage of Y_2_O_3_ spheres, the mixed powder was milled in a horizontal jar mill for 4 h at 100 rpm. Stainless steel balls were used in ball-milling, and the ball-to-powder ratio was 10:1. The well-mixed Ag/Y_2_O_3_ powder was calcined at 400 °C for 2 h, then densified by hot-pressing at 750 °C and 55 MPa for 1 h in an argon atmosphere. Ag/Y_2_O_3_ sintered compacts were thereby obtained.

### 2.4. Characterization

X-ray diffraction (XRD, Smartlab 9, Cu Kα, λ = 1.5406 Å) was utilized to determine the phase, with a scanning rate of 4° 2θ/min. Microstructures were characterized by scanning electron microscopy (SEM, JSM-7001F). Image J software was used to analyze the particles’ size. Samples of ϕ14 mm × 5 mm were machined from the sintered compacts for physical property tests. The density was measured using Archimedes’ method. Vickers hardness tester (401MVD) was adopted to evaluate the hardness at a load of 100 g for 10 s. The electrical conductivity was detected by a vortex conductivity apparatus (FQR7501). Values of hardness and electrical conductivity are statistics based on at least 10 readings for each sample.

The sintered compacts were machined into electrodes (ϕ3 mm × 1.5 mm) for electrical contact tests. The test apparatus (Appendix A) was set up to simulate the switching operation at DC 24 V/10 A. The contact gap between the two electrodes was 10 mm, and the electrodes were contacted for 10^5^ times at the frequency of 1 Hz. After each 5000-switching operation, the contact resistance between two electrodes was detected by the Kelvin four-terminal sensing method (Appendix A). The failure occurred once the value of contact resistance was larger than 10 mΩ [31]. The mass changes were recorded as well.

## 3. Results and Discussion 

### 3.1. Phase, Microstructure, and Physical Properties

Y_2_O_3_ spheres have a cubic structure, which can be indexed to JCPDS No. 41-1105 [32], as shown in Figure 2. Ag/Y_2_O_3_ mixed powder and sintered body show diffraction patterns of both Ag and Y_2_O_3_. The diffraction pattern of Ag is indexed to JCPDS No. 04-0783 [33]. No other crystalline phases were observed.

Figure 3 shows the SEM morphologies and size distribution of spherical Y_2_O_3_ powders. Spherical Y_2_O_3_ powders with narrow size distribution were obtained. The mean sizes of the four powders were 243, 387, 792, and 980 nm, respectively. Particles were individually separated. No aggregation, coalescence, or sintering occurred. After thoroughly mixing with silver powder, Y_2_O_3_ particles retained the spherical shape and good dispersibility, as shown in Appendix A. It can be seen that Ag and Y_2_O_3_ particles are well mixed, which means the sintered body would have a uniform microstructure. 

Figure 4 shows the microstructure of Ag/Y_2_O_3_ sintered compacts. The light area represents the silver phase, while the dark zone is the Y_2_O_3_ phase. After hot-press sintering, Y_2_O_3_ retained the spherical shape and formed a quasi-continuous network structure in the Ag matrix. The network structure is different due to the differences in the size of Y_2_O_3_. The main difference is the distance between the adjacent Y_2_O_3_ particles, which decreases with the size of Y_2_O_3_. Among the four samples, the sintered compact using 243 nm Y_2_O_3_ particles as the raw material has the shortest distance between the adjacent Y_2_O_3_ particles. The unique quasi-continuous network structure would significantly improve electrical performances.

Ag and Y_2_O_3_ powders were firstly mixed in ethanol solution under ultrasonic agitation in our work. Ultrasonic stirring could disperse and activate Ag and Y_2_O_3_ particles. Meanwhile, Y_2_O_3_ spheres could be adsorbed on the silver surface by Van der Waals force and Coulomb force [34]. After drying the ethanol solution, the obtained powder was further mixed by hand grinding and ball milling to avoid the delamination caused by density differences. Then, annealing was conducted to remove the organics and other volatile purities. Finally, the well-mixed Ag/Y_2_O_3_ powder was densified by hot-pressing, and the quasi-continuous network structure was thus formed in the sintered body in a micro-scale. It is difficult to achieve the microstructure refinement and homogenization by a low-energy ball milling for the soft metal matrix composite due to the superplasticity of metal particles [27,28,35,36]. However, both Ag and Y_2_O_3_ phases remained their original size and shape in our sintered compacts to a large extent. Thus, our method enables the design and regulation of the material microstructure in the soft metal matrix composite, facilitating further study on the structure–property relationship.

The mechanical and electrical properties of Ag/Y_2_O_3_ sintered compacts are listed in Table 1. Although no voids were observed in the SEM analysis, all samples’ relative densities are below 100%. This result can be due to the incomplete sintering of Y_2_O_3_ particles. It is well known that the densification temperature of Y_2_O_3_ is above 1900 °C [17,37]. However, the hot-pressing temperature in this work is only 750 °C, which is too low to densify Y_2_O_3_ particles. All sintered bodies exhibit high hardness and high conductivity. The highest conductivity is up to 73.0 IACS%.

### 3.2. Electrical Contact Performances of Ag/Y_2_O_3_ Contacts

Figure 5 shows the variation in contact resistance (*R*_c_) and mass of Ag/Y_2_O_3_ contacts during the electrical contact test. The contact resistances were low and stable during the whole test for the contacts prepared using 243 nm and 387 nm Y_2_O_3_. However, intermittent failure (*R*_c_ > 10 mΩ) occurred after 5 × 10^4^ switching operations for the other two contacts. The mass changes were also different between the four contacts and divided into several stages, as shown in Figure 5(a_1_–d_1_). In stage I and stage II, the material transfer direction was from the cathode to the anode. Thus, the mass of the anode increased as the cycle number increased, while the mass of the cathode decreased. Stage III was found for the contacts prepared using 792 nm and 980 nm Y_2_O_3_. In stage III, the mass of both anode and cathode decreased as the cycle number increased, and the failure occurred after a sudden mass drop of the anode. According to the theory of electrical contacts [38,39], the electric current between the anode and cathode is achieved through the interface’s metal-to-metal spots (α-spots). The mass abruptly decreased once the spalling occurred, which produced a sudden decrease in α-spots. Simultaneously, the electrode surfaces were severely worsened, increasing the contact resistance sharply. The analysis of the worsened microstructure will be shown later. In all, the contacts prepared using 243 nm and 387 nm spherical Y_2_O_3_ exhibited outstanding electrical contact performance measures, such as low contact resistance, stable mass transfer from the cathode to anode, and no failure up to 10^5^ cycles tests.

The morphologies of cathode and anode after 10^5^ cycle tests are shown in Figure 6. The contact prepared using 243 nm spherical Y_2_O_3_ shows relatively uniform erosion across the whole surface. Corrosion pits are observed on both the cathode and anode. As for the surface of contact prepared using 387 nm Y_2_O_3_, most parts show relatively uniform erosion, but a crater and protrusion appear, as shown in Figure 6b,b_1_. The other two contacts show more severe erosion on both the cathode and anode. These results agree well with the change in the contact resistance and mass shown in Figure 5. Research [40,41] has pointed out that a relatively uniform and smooth surface means more contact spots, thereby creating more electron transfer paths and reducing the contact resistance.

### 3.3. Microstructure Change after One-Time Arcing Erosion

As the anode approaches or leaves the cathode, electrons generate and escape from the cathode to the anode under an electrical field. During this moving process, electrons bombard gas molecules and metal vapors, producing large amounts of positive ions and electrons. An arc will form once the charged particles reach saturation (Appendix A) [1]. Then, the positive ions bombard the cathode, resulting in the jet of the cathode material. These materials are transferred to the anode along with the electron flow [42]. The crater and protrusion are the products of this material transfer process (Appendix A). 

The failure of contacts is the result of the gradual worsening of the microstructure. Figure 4 shows the original surface microstructure of Ag/Y_2_O_3_ electrical contacts. Spherical Y_2_O_3_ particles were distributed in the Ag matrix without pits or pores. However, as shown in Figure 7, an erosion crater (~500 μm) appeared on the cathode after one-time arcing erosion. In the erosion zone, the surface became rough and was covered by smaller ion bombardment craters (<10 μm). For the cathodes prepared with the Y_2_O_3_ of 243 nm and 387 nm, these smaller craters connected and formed a continuous network. In contrast, the other two contacts show isolated and deep erosion pits. The microstructure differences determine the mass loss. Generally, the continuous structure indicates rapid arc motion, which may decrease the mass loss. Instead, the isolated and deep erosion pits indicate slow arc motion or even stationary arcing, thereby accelerating the mass loss [5,43]. The results shown in Figure 5 confirmed this relationship.

As mentioned above, the mass transferred from the cathode to the anode upon the arc erosion. Therefore, the anode exhibited a different microstructure change in contrast to the cathode. A protrusion with isolated and larger pores was observed, as shown in Figure 8. Backscattered electron (BSE) images exhibit mainly two phase zones. The bright zone indicates the Ag-rich phase, while the dark area indicates the Y_2_O_3_-rich phase. Noticeably, the anodes prepared using 792 nm and 980 nm Y_2_O_3_ show coarsened Ag-rich zones and cracks. As mentioned above, the material is transferred from the cathode and deposited on the anode upon the arcing erosion. Meanwhile, the hot electrons bombard the anode, leading to the melting of the anode and thus forming a molten pool. The low-density oxide spheres would float upward and redistribute [44], thereby creating the structure shown in Figure 8(a_3_–d_3_). Due to the thermal stress, the cracks formed and propagated in the Y_2_O_3_-rich zone. Once the cracks propagated to some extent, spalling occurred, and the mass reduced sharply. Thereby, the failure occurred, as shown in Figure 5. In contrast, the anodes prepared using 243 nm and 387 nm Y_2_O_3_ exhibit a relatively uniform microstructure with no coarsened Ag-rich zones. Besides, no cracks are observed. These factors account for the outstanding electrical performances measures, such as the low contact resistance and lower mass loss shown in Figure 5.

### 3.4. Mechanism Analysis

From the viewpoint of electrical performances and microstructure, the contact prepared using 243 nm Y_2_O_3_ is the best among the four contacts. The differences in Y_2_O_3_ size led to the different primitive microstructures. These microstructures changed differently upon arcing erosion. Therefore, the primitive microstructure of contacts is the key factor to determine the subsequent changes in structure and electrical performance.

Y_2_O_3_ has a lower work function when compared with Ag [45]. Guan et al. [43,46] noted that the arc movement depends on the distance between the particles of lower work function. Therefore, the Y_2_O_3_ particles on the cathode could act as the arc moving sites based on the rule of arc motion. The arc could move along the Y_2_O_3_ network if the Y_2_O_3_ spacing is small enough, as shown in Figure 9a. If Y_2_O_3_ particles are isolated, as shown in Figure 9b, the arc stays around the individual Y_2_O_3_ particle, thereby burning the zone around the Y_2_O_3_ particle.

As shown in Figure 4, the contact prepared using 243 nm Y_2_O_3_ exhibits a quasi-continuous network structure. In this structure, Y_2_O_3_ particles are so close to each other that the arc could move along the Y_2_O_3_ network. During the rapid moving process, the arc energy releases quickly before extinguishing. Eventually, the shallow and connected erosion craters formed, as shown in Figure 7. If the distances between Y_2_O_3_ particles are too far to jump over, the arc would stay and burn the zone around the individual Y_2_O_3_ particle, thereby forming isolated and deep erosion pits. Guo et al. [47] pointed out that the fine particles could improve the anti-arc performance by increasing the viscosity of the molten pool. Our work provided a new explanation of particle-size effects on electrical contact performance from the perspective of arc movement.

## 4. Conclusions

Ag/Y_2_O_3_ electrical contacts were successfully obtained based on a low-energy ball milling treatment. Spherical Y_2_O_3_ particles were uniformly distributed in the Ag matrix. Changes in Y_2_O_3_ particle size at the submicron scale had a remarkable influence on the contacts’ electrical performance. The Ag/Y_2_O_3_ contact prepared using 243 nm Y_2_O_3_ exhibited the most outstanding performance, including low contact resistance, less mass loss, and long lifetime. The formed quasi-continuous network structure was responsible for the outstanding electrical performances, which significantly improved the anti-arc ability of the electrical contacts.

## Figures and Tables

**Figure 1 materials-15-02450-f001:**
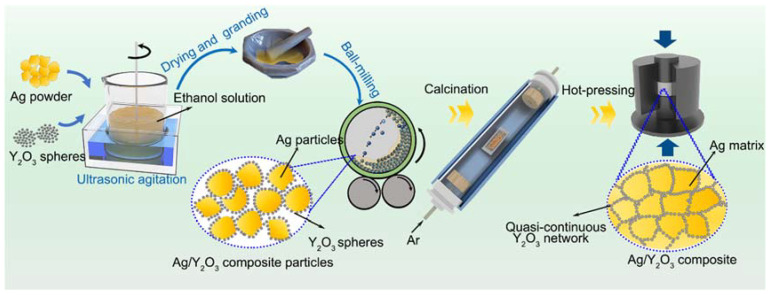
Schematic diagram of the preparation of Ag/Y_2_O_3_ sintered compact.

**Figure 2 materials-15-02450-f002:**
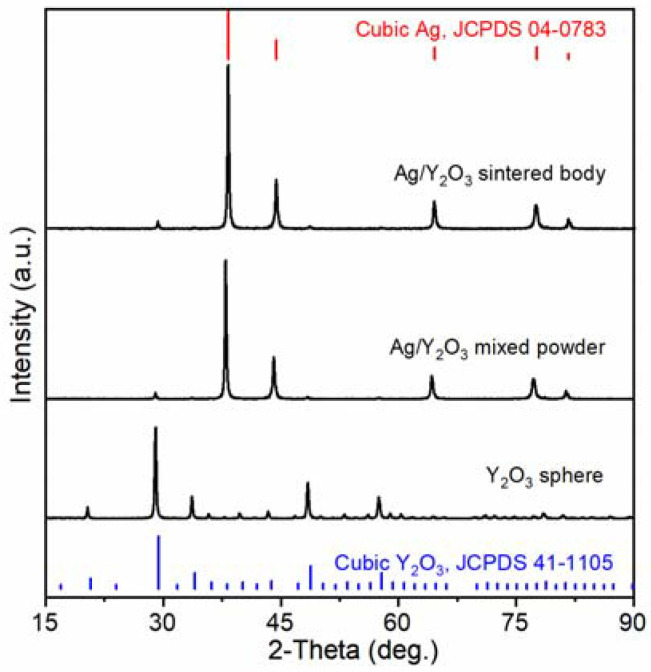
XRD patterns of Y_2_O_3_ spheres, Ag/Y_2_O_3_ mixed powder, and Ag/Y_2_O_3_ sintered body.

**Figure 3 materials-15-02450-f003:**
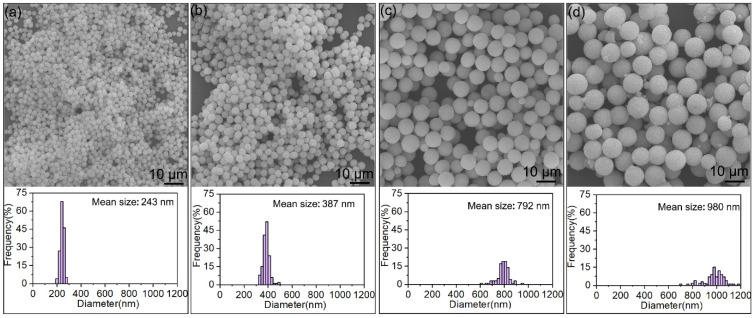
SEM images and size distribution of spherical Y_2_O_3_ powders: (**a**) 243 nm; (**b**) 387 nm; (**c**) 792 nm; (**d**) 980 nm.

**Figure 4 materials-15-02450-f004:**
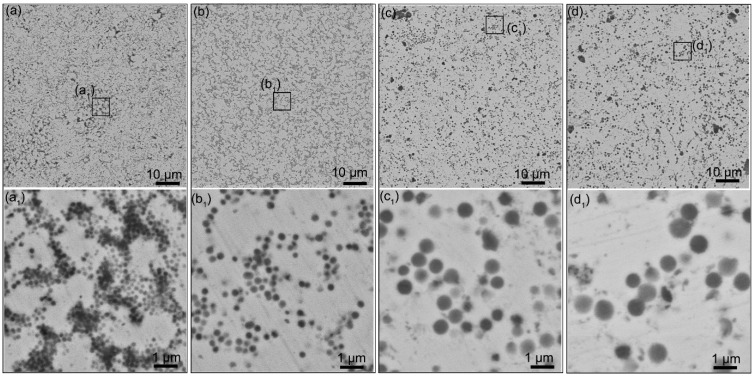
(**a**–**d**) Low- and (**a_1_**–**d_1_**) high-magnification SEM images of Ag/Y_2_O_3_ sintered compacts. The mean size of Y_2_O_3_ used: (**a**,**a_1_**) 243 nm, (**b**,**b_1_**) 387 nm, (**c**,**c_1_**) 792 nm, (**d**,**d_1_**) 980 nm.

**Figure 5 materials-15-02450-f005:**
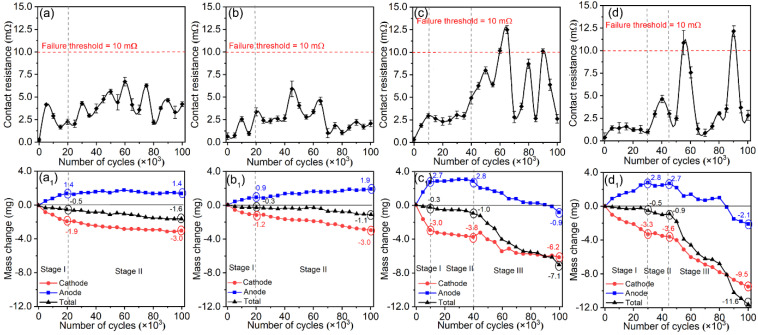
The change in contact resistance (**a**–**d**) and mass (**a_1_**–**d_1_**) as the cycle number of contact tests increased. The mean size of Y_2_O_3_ used: (**a**,**a_1_**) 243 nm, (**b**,**b_1_**) 387 nm, (**c**,**c_1_**) 792 nm, (**d**,**d_1_**) 980 nm.

**Figure 6 materials-15-02450-f006:**
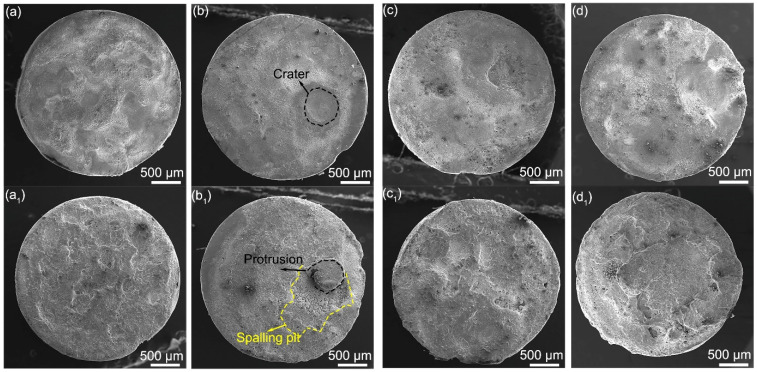
The morphologies of cathode (**a**–**d**) and anode (**a_1_**–**d_1_**) after 10^5^ cycle tests. The mean size of Y_2_O_3_ used: (**a**,**a_1_**) 243 nm, (**b**,**b_1_**) 387 nm, (**c**,**c_1_**) 792 nm, (**d**,**d_1_**) 980 nm.

**Figure 7 materials-15-02450-f007:**
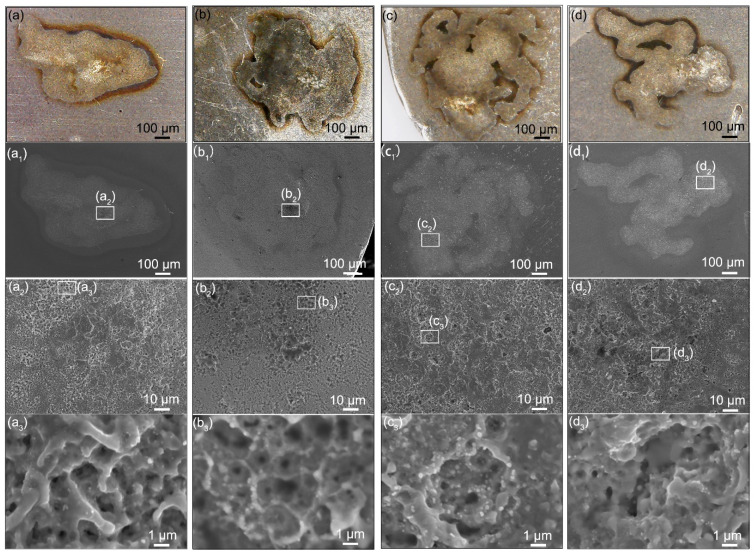
Microstructure change of the Ag/Y_2_O_3_ cathodes after one-time arcing erosion: (**a**–**d**) optical images, (**a_1_**–**d_1_**) low-magnification SEM images, (**a_2_**–**d_2_**,**a_3_**–**d_3_**) high-magnification SEM images. The mean size of Y_2_O_3_ used: (**a**–**a_3_**) 243 nm, (**b**–**b_3_**) 387 nm, (**c**–**c_3_**) 792 nm, (**d**–**d_3_**) 980 nm.

**Figure 8 materials-15-02450-f008:**
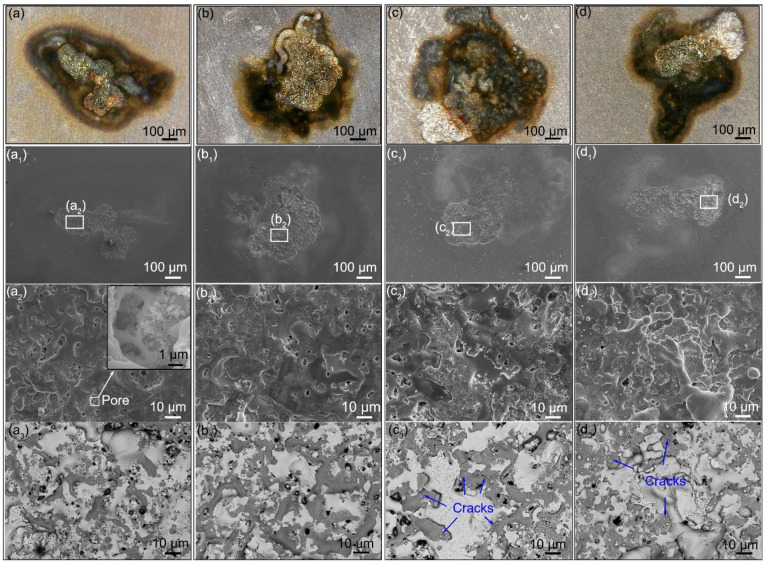
Microstructure changes of the Ag/Y_2_O_3_ anodes after one-time arcing erosion: (**a**–**d**) optical images, (**a_1_**–**d_1_**) low-magnification SEM images, (**a_2_**–**d_2_**) high-magnification SEM images, and (**a_3_**–**d_3_**) backscattered electron (BSE) images. The mean size of Y_2_O_3_ used: (**a**–**a_3_**) 243 nm, (**b**–**b_3_**) 387 nm, (**c**–**c_3_**) 792 nm, (**d**–**d_3_**) 980 nm.

**Figure 9 materials-15-02450-f009:**
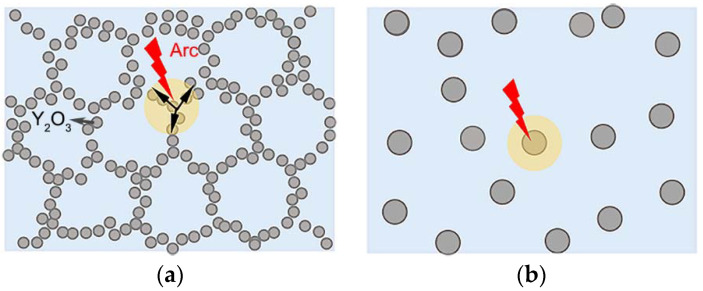
Arc movement characteristics: (**a**) moving along the Y_2_O_3_ network, (**b**) staying around the Y_2_O_3_ particles.

**Table 1 materials-15-02450-t001:** Physical properties of Ag/Y_2_O_3_ sintered compacts.

Y_2_O_3_ Mean Size (nm)	Relative Density (%)	Hardness (Hv0.1)	Conductivity (IACS%)
243	96.8	107.2 ± 3.9	67.6
387	98.1	109.2 ± 1.8	73.0
792	96.1	105.8 ± 3.4	69.9
980	94.9	103.7 ± 2.1	66.7

## Data Availability

The data presented in this study are available from the corresponding authors upon reasonable request.

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
