# Peer review of "Quasi-Continuous Network Structure Greatly Improved the Anti-Arc-Erosion Capability of Ag/Y2O3 Electrical Contacts"

_materials, 2022, doi:10.3390/ma15072450_

Round 1

Reviewer 1 Report

The manuscript can be accepted with minor revisions.
1) In the introduction part, explain briefly why metal oxide is needed for electrical contact . Is Ag alone not enough? Can metal oxide be replaced with other materials?  
2) Make it more clear on the Y2O3 powder preparation. For example 1:33 and 1:133 ratios can obtain the particle size in between 200 to 400 nm. Or, please state clearly in section 3.1 that the powder with 243, 387, 792 and 980 nm belong to which ratio?
3) 243 nm shows the best performance. Do you think if you go to a smaller size than 243 nm would the results will be better?
4) How are you going to optimize the perfect average size for the Y2O3?  

Author Response

Please see the attachment named "Response to Reviewer 1 Comments.docx".

Reviewer 2 Report

My comments were added to the "Word.docx" file as added notes that they should be addressed.

Author Response

Please see the attachment named "Response to Reviewer 2 Comments.docx"

Round 2

Reviewer 2 Report

The authors corrected the original version of the manuscript.
I accept this work for publication in the present form.